# Carrot Pomace Polysaccharide (CPP) Improves Influenza Vaccine Efficacy in Immunosuppressed Mice via Dendritic Cell Activation

**DOI:** 10.3390/nu12092740

**Published:** 2020-09-09

**Authors:** Pureum Sun, Yeeun Kim, Hoyoung Lee, Jihyun Kim, Bok Kyung Han, Eunbyeol Go, Somin Kwon, Ju-Gyeong Kang, Sooseong You, Jaeyul Kwon

**Affiliations:** 1Department of Infection Biology, College of Medicine, Chungnam National University, 266 Munhwa-ro, Jung-gu, Daejeon 35015, Korea; reum6133@naver.com (P.S.); dpdms1189@hanmail.net (Y.K.); geb0816@gmail.com (E.G.); 2Clinical Medicine Division, Korea Institute of Oriental Medicine, Daejeon 34054, Korea; hylee@corepharma.co.kr (H.L.); kimjh763@kiom.re.kr (J.K.); 3Department of Food and Biotechnology, Korea University, Sejong 30019, Korea; hanmoo@korea.ac.kr; 4Laboratory of Neurogenetics, National Institute on Aging, NIH, Bethesda, MD 20892, USA; kwon.somin@gmail.com; 5Cardiovascular Branch, National Heart, Lung, and Blood Institute, NIH, Bethesda, MD 20892, USA; kangju@mail.nih.gov; 6Department of Medical Education, School of Medicine, Chungnam National University, Daejeon 35015, Korea; 7Translational Immunology Institute, Chungnam National University, Daejeon 35015, Korea

**Keywords:** BMDC, dendritic cells, carrot pomace polysaccharides, influenza vaccine, immunosuppression, innate immune cells, cyclophosphamide, inflammatory cytokines

## Abstract

Despite the advancements in vaccination research and practices, influenza viruses remain a global health concern. Inducing a robust immune response by vaccination is especially challenging in the elderly, the immunocompromised, and persons with chronic illnesses. Polysaccharides derived from food may act as a safe and readily accessible means to boost the immune system during vaccination. In this study, we investigated whether crude polysaccharides derived from carrot pomace (CPP) could stimulate innate immune cell function and promote influenza vaccine immunogenicity. In bone marrow-derived dendritic cells (BMDCs), CPP increased the fraction of CD11c+MHCII+ cells and the expression of co-stimulatory molecules CD40 and CD80, indicative of enhanced maturation and activation. Functionally, CPP-treated BMDCs promoted inflammatory cytokine production in splenic lymphocytes. In a mouse model of immunosuppression induced by cyclophosphamide, animals given CPP before and after an influenza vaccine challenge showed increased frequencies of dendritic cells and natural killer cells in the spleen, in addition to the recovery of vaccine-specific antibody titers. Moreover, innate myeloid cells in CPP-fed mice showed evidence of phenotypic modification via markedly enhanced interleukin(IL)-12 and interferon(IFN)-γ production in response to lipopolysaccharide(LPS) stimulation ex vivo. Our findings suggest that the administration of carrot pomace polysaccharides can significantly enhance the efficacy of influenza vaccination.

## 1. Introduction

The influenza virus is among the most common causes of human respiratory infections, leading to millions of severe cases worldwide and up to 500,000 deaths every year [1]. Despite the scientific advances to improve vaccine efficacy, standard inactivated influenza vaccines fail to elicit an adequate response in populations with underdeveloped or impaired immune systems, particularly in newborns, the elderly, and immunocompromised individuals such as transplant recipients and chemotherapy patients [2,3,4,5]. These populations are not only vulnerable to infection, but also often have a high risk of influenza-related morbidity, contraindicating the use of live attenuated vaccines that may have induced a more robust response.

Strategies to enhance inactivated vaccine immunogenicity include using higher doses of antigen, as is currently recommended for the elderly, or co-administering an appropriate adjuvant. Adjuvants may enhance immunogenicity and increase the duration of protection, while also promoting antigen sparing [6,7]. Very few adjuvants have been approved for use, however, compiling the necessary proof of safety is a rigorous and stringent process that can take many years and delay vaccine development [8]. As such, developing efficient, accessible, and inherently safe strategies to improve host response to vaccination are highly warranted.

Plant polysaccharides, including mannan, delta inulin, β-glucan, starch, dextran, and pectin, are receiving attention as attractive candidates for vaccine adjuvants due to their intrinsic immunomodulating properties, accessibility, and low toxicity [9]. Furthermore, genetic engineering approaches may allow for low production costs and even facilitate oral delivery. In animal infection models, polysaccharides from Panax ginseng polysaccharide were shown to be protective against H1N1 and H3N2 influenza viruses [10], and the Eupatorium adenophorum polysaccharide improved the protective efficacy of the H5N1 vaccine [11]. More recently, mannan conjugated to a whole inactive H1N1 influenza virus was reported to elicit higher serum IgG than immunization with the virus alone [12]. Starch made into the form of “bioneedles” and injected into the recipient along with the lyophilized vaccine is also being explored as an alternative to conventional vaccination [13].

In this study, we investigated how the oral delivery of polysaccharides derived from carrot pomace (CPP) can impact influenza vaccine efficacy. Food-derived polysaccharides may be valuable sources for nonconventional adjuvants, as they are safe to consume, may be more acceptable to the public, and many are already purported to boost host immunity [14,15,16]. Given that the mechanism of an effective oral adjuvant would likely be independent of direct association with the antigen, we examined the effects of CPP based on phenotypic changes in innate immune cells. Using blood-derived dendritic cell cultures and an animal model of immunosuppression, we show that CPP may promote the maturation, expansion, and function of antigen-presenting innate immune cells to boost influenza vaccine efficacy.

## 2. Materials and Methods

### 2.1. Mice

Animal experimental procedures were approved by the Institutional Animal Care and Use Committee (IACUC) at the Korean Institute of Oriental Medicine (17-070) and Chungnam National University (CNU-00899). Bone marrow cells were obtained from 6 to 10 week-old female C57BL/6 mice (Damul Science, Daejeon, Korea). For the influenza vaccine experiment, 7 to 8 week-old BALB/c female mice were obtained from Oriental-Bio Co. (Seongnam, Korea) and housed under pathogen-free conditions with freely available food and water.

### 2.2. Influenza Vaccine

Mice were vaccinated with inactivated Fluzone^®^ 2017/2018 influenza vaccine (Seoul, Korea) stocks containing four different strains (30 µg/mL): A/Michigan/45/2005(H1N1)pdm09-like strain, A/Hong Kong/4801/2014(H3N2)-like strain, B/Phuket/3073/2013-like strain and B/Brisban/60/2008-like strain.

### 2.3. Immunization

Mice were administered intraperitoneally (IP) with saline or cyclophosphamide (CTX, 150 mg/kg mouse body weight, Sigma-Aldrich, St. Louis, MO, USA) three times every other day. At 13 days following the last CTX injection, mice were immunized intramuscularly with the Fluzone^®^ 2017/2018 influenza vaccine (1 µg of each strain/mouse). Polysaccharide or control solutions were given daily for 10 or 20 days. Mice in control groups were orally administered 0.25% carboxymethyl cellulose (CMC)(Sigma-Aldrich, St. Louis, MO, USA) or green tangerine polysaccharides (GTP) (300 mg/kg/day) 10 days before and after inoculation (total 20 days). Mice in experimental groups were orally administered CPP (300 mg/kg/day or 600 mg/kg/day) 10 days prior to inoculation (total 10 days) or 10 days before and after inoculation (total 20 days). Serum and splenocytes were collected at 4 weeks after immunization to measure influenza-specific antibody titers and flow cytometric analysis, respectively.

### 2.4. Preparation of Food-Derived Polysaccharides

Carrot pomace polysaccharides (CPP) and green tangerine polysaccharides (GTP) were provided in a powder form from BKbio Co., Ltd. (Jeju, Korea). CPP and GTP were prepared as in Park et al. [17]. Briefly, green tangerine peels or carrot pomace were dried in a conventional oven for 24 h, chopped, suspended in 20 volumes of distilled water, and treated with pectinase (Pectinex^®^ Ultra SP-L from Aspergillus aculeatus, Novozymes A/S, Krogshøjvej, Denmark) for 6 h at 50 °C. After enzyme deactivation by heating at 85 °C for 15 min, the supernatant was collected by centrifugation (2500× *g*, 20 min). Four volumes of 95% ethanol were added to the supernatant and stirred slowly to allow precipitation to form overnight. The precipitates were dissolved in distilled water and a substance with a molecular weight of 10,000 Da or less was extracted by ultrafiltration (Pellicon^®^ 2 Ultrafiltration Cassettes, Merck KGaA, Darmstadt, Germany). The retentate was lyophilized for use in experiments. The yield of CPP was 1–2% (*w*/*w*) from carrot and 3–5% from carrot pomace.

### 2.5. Bone Marrow-Derived Dendritic Cell (BMDC) Culture

Bone marrow cells were obtained from the femur of 6 to 10 week old female C57BL/6 mice and differentiated to dendritic cells as previously described [18]. Briefly, bone marrow cells were cultured in RPMI medium containing 10% FBS, 1% penicillin/streptomycin, 50 μM 2-Mercaptoethanol, and 20 ng/mL GM-CSF for 3 days. The medium was replaced with fresh supplemented medium and cultured for an additional 3 days. Nonadherent immature dendritic cells were harvested on day 6, treated with 100 to 1000 μg/mL polysaccharide or vehicle in complete media for 24 h, and subjected to flow cytometry analysis.

### 2.6. Mixed Lymphocyte Reaction

Allogenic mixed lymphocyte reactions were performed as previously described [19]. In brief, BMDCs prepared from C57BL/6 mice were incubated with food-derived polysaccharides or controls for 1 day, washed with complete medium, and incubated with splenocytes from BALB/c mice for 3 days. The culture supernatant was used for cytokine analysis.

### 2.7. Cytokine Beads Array

Cytokines in the supernatants of mixed lymphocyte reactions were measured with the BD CBA Mouse Th1/Th2/Th17 Cytokine Kit (BD Bioscience, San Jose, CA, USA) according to the manufacturer’s instructions.

### 2.8. Cell Staining and Flow Cytometry

Cells were stained with conjugated antibodies as follows and analyzed with a flow cytometer (FACS Canto II, BD Biosciences, San Jose, CA, USA) and FlowJo software: PerCP-Cy5.5-MHCⅡ (562363, BD Biosciences, San Jose, CA, USA), PE-Cy7-anti-CD80 (25-0801-82, eBioscience, San Diego, CA, USA), APC-anti-CD11c (20-0114-U100, Tonbo bioscience, San Diego, CA, USA ), APC-Cy7-CD11b (557657, BD Biosciences, San Jose, CA, USA). Cell viability was measured by LIVE/DEAD Fixable Violet dye (L34955, Thermo Fisher Scientific, Waltham, MA, USA). To assess ex vivo IL-12 or IFN-γ expression of innate immune cells in the animals, the spleens were harvested at 4 weeks after viral vaccination. Splenocytes were prepared and treated with 1 μg/mL LPS (L6529, Sigma-Aldrich, St. Louis, MO, USA) in a medium containing 10 μg/mL Brefeldin A (BFA) (Sigma-Aldrich, St. Louis, MO, USA). The cells were stained with the following antibodies for 30 min at 4 °C: PE-conjugated anti-NK1.1 (557391, BD Biosciences, San Jose, CA, USA), PerCP-Cy5.5-conjugated anti-MHC-II (562363, BD Biosciences, San Jose, CA, USA), APC-conjugated anti-CD11c (20-0114-U100, TONBO), APC-H7-conjugated anti-CD11b (557657, BD Biosciences, San Jose, CA, USA), and BV421-conjugated anti-F4/80(123137, Biolegend, San Diego, CA, USA) antibodies. Intracellular cytokine staining was performed using BD Cytofix/Cytoperm (554714, BD Biosciences, San Jose, CA, USA) and BD Perm/Wash buffers (554723, BD Biosciences, San Jose, CA, USA). Intracellular IL-12 was detected by FITC-conjugated IL12p40/p70-specific antibody (560564, BD Biosciences, San Jose, CA, USA). IFN-γ was detected by PE-Cy7-conjugated IFN-γ-specific antibody (557649, BD Biosciences, San Jose, CA, USA). Mouse Fc receptors were blocked with Mouse Fc Block™ (553142, BD Biosciences, San Jose, CA, USA) for 15 min at 4 °C.

### 2.9. Influenza Antigen-Specific Antibody Titer Measurement

Influenza antigen-specific antibodies in mouse serum were titrated with enzyme-linked immunosorbent assay (ELISA). Briefly, the plates were coated with 4 μg/mL Fluzone^®^ 2017/2018 influenza vaccine (1 μg/mL each Influenza antigens), incubated with 10000x diluted mouse serum after blocking with 5% bovine serum albumin (BSA) in phosphate-buffered saline (PBS), washed, and subsequently incubated with horseradish peroxidase(HRP)-conjugated anti-IgG (1:10000; Abcam, Cambridge, MA, USA), anti-IgG1 antibody (1:10000; Abcam), or anti-IgG2a antibody (1:10000; Abcam). After sufficient color development of the substrate solution, absorbance was measured at 405 nm by a microplate reader.

### 2.10. Statistical Analysis

Unless otherwise specified, data are shown as the mean ± SD and each experiment was repeated two or three times. Data were analyzed by the two-tailed unpaired t-test or one-way ANOVA with Tukey’s post hoc analysis using GraphPad Prism (v7.02, GraphPad, La Jolla, CA, USA).

## 3. Results

### 3.1. Bone Marrow-Derived Dendritic Cell (BMDC) Maturation and Activation In Vitro Was Promoted by Food-Derived Polysaccharides

Given that dendritic cells (DCs) are critical for the initiation of innate and adaptive immune responses, we investigated the effects of food-derived polysaccharides on GM-CSF-driven DC maturation. In brief, bone marrow cultures were differentiated with GM-CSF, treated with food-derived polysaccharides, lipopolysaccharide (LPS), or vehicle, and measured for cell surface marker expression by flow cytometry. It has been reported that LPS treatment increases the expression of the costimulatory molecules and induces the maturation of DCs [17]. As such, LPS was used as a positive control that can induce DC maturation in vitro. In addition, green tangerine polysaccharide (GTP), known to enhance the production of IL-6, TNF-α, and nitric oxide (NO) in macrophage cell lines, was used as a positive control of immune-boosting food-derived polysaccharides [18]. CPP treatment up to 1000 μg/mL caused no significant cytotoxic effect in the BMDCs, whereas 100–200 μg/mL of GTP reduced cell viability (Figure 1a). The expression of CD11c (α integrin), a classic marker of myeloid dendritic cells, was unchanged in BMDCs treated with either polysaccharide. However, the fraction of mature DCs, as indicated by CD11c+ cells co-expressing MHCII+, was increased by both CPP and GTP (Figure 1a). In addition, 400 μg/mL CPP increased the expression of co-stimulatory molecules such as CD40 and CD80 in a dose-dependent manner, suggestive of enhanced DC activation (Figure 1b). A lower dose of GTP (100 μg/mL) also showed a strong effect on CD40 and CD80.

To assess whether these phenotypic changes translated into enhanced functionality, namely the initiation of the adaptive immune response, we performed a mixed (allogeneic) lymphocyte reaction (MLR). The MLR is a standard assay widely used to measure antigen presenting-cell activity. By mixing stimulator antigen-presenting cells from one strain with responder cells from another, the responder cells (usually T cells) can recognize the allogeneic stimulator cells as foreign and undergo potent activation. Subsequent cytokine levels can then be detected as a measure of lymphocyte activation and thereby provide indication of the antigen-presenting capacity of stimulator cells. Briefly, GM-CSF-derived BMDCs from C57BL/6 mice were used as stimulator cells and splenocytes from BALB/c mice as responder cells. BMDCs were treated with CPP, GTP, or vehicle for one day, washed, and co-cultured with the BALB/c splenocytes for 3 days. It is of note that basal cytokine production in the splenocytes + BMDC mixed reaction (lane 3, IL-17; 0.039, IL-6; 0.69 ng/mL) is a few times higher than those of BMDCs only (lane 1; IL-17; 0.003, IL-6; 0.11 ng/mL), indicating that the primary source of these cytokines are likely splenocytes (Figure 1c). The resulting medium was analyzed by capture bead assay for cytokine secretion. Remarkably, CPP-treated BMDCs induced a significant increase in the production of IL-6, TNFα, IL-17 and IL-10 when compared to the vehicle-treated group (Figure 1c). These results suggest that CPP treatment promotes BMDC maturation and lymphocyte-activation capacity.

### 3.2. CPP Increases Dendritic Cell and Natural Killer(NK) Cell Population in Mouse Splenocytes

Based on the above in vitro observations, suggesting that CPP can enhance the maturation and activity of DCs, we sought to determine whether CPP could boost an immunosuppressed animal’s immune response to an inactivated influenza vaccine. To induce immunosuppression, mice were treated with cyclophosphamide (CTX), an alkylating agent well known to rapidly deplete neutrophils and proliferating lymphocytes [19]. Mice were given IP injections of CTX for five days, then orally administered CPP, GTP, or control solutions for 10 days before (total 10 days) or both before and after (total 20 days) receiving an influenza vaccine (Figure 2a). The flow cytometry analysis of isolated mouse splenocytes revealed that 20 days of CPP treatment increased the percentage of CD11c+MHCII+ DCs to a level comparable to that of GTP treatment (Figure 2b,c). The population of CD11b+NK1.1+ natural killer cells in the spleen was also approximately 2.5 times higher in CPP- than in vehicle-treated mice (Figure 2d). Interestingly, CPP given 10 days prior to vaccination did not affect the number of immune cells, suggesting that oral polysaccharides may be the most effective when given simultaneously to or following vaccination. These data suggest that innate immune cell expansion, such as that of DCs and NK cells, are sensitive to food-derived polysaccharides.

### 3.3. CPP Enhances the IL-12 Production Ability of Innate Immune Cells

An important modulator of the antiviral immune response is cytokine release by innate immune cells. IL-12 is particularly important for facilitating a coordinated immune response, as it is known to enhance the cytotoxic activity of NK cells and CD8+ T cells, as well as stimulate naïve T cells to differentiate into Th1 cells [20,21,22,23]. To examine whether CPP impacts IL-12 production in innate immune cells, we used flow cytometry to measure the fraction of IL-12+ cells in isolated CD11b+ and CD11c+ populations stimulated by LPS (Figure 3). Both populations showed significantly more IL-12+ cells after 20 days of CPP treatment at fractions comparable to that of GTP (Appendix A). To investigate which innate immune cell types were responsible for this enhanced IL-12 production, we performed the same analysis in cell populations enriched in DC (CD11c+MHCII+), NK cells (CD11b+NK1.1+), or macrophages (CD11b+F4/80+) and stimulated with LPS. Remarkably, all three populations showed a striking increase in IL-12+ cells after 20 days of CPP treatment (Figure 3c). These cell types are also important producers of IFN-*γ*, a vital cytokine for protection against viral infection through mechanisms including the inhibition of viral entry, the disruption of viral replication, and the blockade of viral protein translation [24]. CPP treatment indeed impacted IFN-*γ* production, markedly enhancing the proportion of IFN-*γ*-producing cells in the DC- and macrophage-enriched (MP) populations in a dose-dependent manner (Figure 3d). Altogether, these data suggest that CPP may enhance cytokine production in innate immune cells to promote a sustained, anti-viral response.

### 3.4. CPP Treatment Enhanced Antibody Production to Vaccine Challenge

Given that CPP may stimulate the anti-viral activity of the innate immune system, we were interested in the humoral immune response to an influenza vaccine challenge. Consistent with previous literature [25,26], vaccine-specific antibody titers were almost abolished in cyclophosphamide-injected animals given a quadrivalent inactivated influenza vaccine (Figure 4). Remarkably, treatment with CPP significantly recovered the total IgG response to a level comparable to that of GTP. Given that CTX-induced immunosuppression was not fully recovered by CPP treatment, however, further investigation is warranted on CPP’s role in influenza viral vaccine-mediated adaptive immunity (Figure 4a). Subclass-specific antibody concentrations for IgG1 and IgG2 also demonstrated modest recovery, although without reaching statistical significance. These findings suggest that CPP can enhance protection against influenza viral infection.

## 4. Discussion

In this study, we sought to explore a novel influenza vaccination strategy in a mouse model of immunosuppression. Importantly, we showed that oral treatment with carrot pomace-derived polysaccharides (CPP) can partially restore the vaccine-specific total IgG response in immunosuppressed mice. This immune-boosting effect may be attributable to not only a marked expansion of total macrophage, DC, and NK cells in the spleen of CPP-treated mice, but also a dramatic increase in those positive for IL-12 and IFN-γ. In addition, BMDC cultures treated with CPP showed an increased expression of maturation markers and stimulated co-cultured lymphocytes to release elevated levels of cytokines. Altogether, CPP may improve influenza vaccine immunogenicity in immunosuppressed mice by enhancing antigen presentation and cytokine production in innate immune cells.

Dendritic cells are traditionally considered to be the bridge between the two immune responses, presenting antigens to adaptive immune cells and generating copious amounts of cytokines to orchestrate both innate cell activity and adaptive cell differentiation [27,28]. Our data showed that CPP can induce the upregulation of MHC II, CD11c, CD80, and CD40 in BMDC cultures, suggesting greater antigen presentation and co-stimulation functionality [29,30]. It is noteworthy that performing these experiments with cDCs or other dendritic cell types may provide additional information to that of BMDCs [31]. The polysaccharide adjuvant derived from delta inulin, Advax, has been reported to induce similar phenotypic changes in APCs, enhancing antigen presentation to promote cellular and humoral responses specific to a trivalent influenza vaccine [32]. Various other plant-derived polysaccharides have been shown to upregulate DC function, including the active polysaccharide in aloe vera, acemannan [33], and polysaccharides from *Astragalus mongholicus* [34]. Polysaccharides derived from mushroom and barley were also shown to increase HLA-DR, CD40, CD80, and CD86 expression in human DCs [35].

We also observed phenotypic changes in innate immune cells in vivo, where oral CPP significantly increased the percentage of IL-12- and IFN-γ-producing DCs and macrophages in immunosuppressed mice. Given that IL-12 and IFN-γ are critical for the early polarization and sustained effector activity of CD4+ Th1 cells [36], CPP may induce a Th1-skewed response. DCs are also important regulators of NK cells, the representative innate lymphoid cells and first line of defense against virus-infected host cells [37,38]. IL-12 released from DCs is a potent inducer of IFN-γ production in NK cells [39,40]. Although the proportion of IFN-γ-expressing NK cells was unaffected by CPP treatment, the absolute number of such cells was likely increased due to the expansion of the total NK cell population. NK cells also display features of antigen-specific memory, undergoing robust secondary expansion and degranulation in response to exposure to a familiar virus [41,42].

Interestingly, the influenza vaccine alone did not induce NK cell expansion in either control or immunosuppressed mice although it has been previously reported in infections with cytomegalovirus, influenza A, and vaccinia virus [41,43,44]. Only mice treated with CPP before and after vaccination demonstrated heightened innate immune cell activation. Although the underlying mechanism requires further study, our results indicate that unlike alum, CPP does not need to be associated with the antigen to boost immunogenicity. Like Advax, however, simply priming local APCs prior to antigen presentation is insufficient [32]. The effects of CPP may be related to a relatively new concept termed “trained immunity”, which describes the innate immune cells that produce a heightened, secondary response to encounters with the same or novel pathogen up to several months after the initial exposure [45]. In our study, the innate immune cells from mice that received CPP treatment before and after vaccination mounted a significant cytokine response to an LPS challenge, suggestive of trained immunity. Furthermore, the significant expansion and activation of myeloid and NK cells in CPP-treated mice was observed at four weeks post-vaccine challenge, even though pro-inflammatory transcriptional programs in innate immune cells are thought to only last in the order of days. A recent study in non-human primates observed similarly late phenotypic modifications, such as increased an expression of maturation and activation markers, in innate immune cells between 2 weeks and 2 months post vaccination with modified Vaccinia Ankara [46]. In another study, circulating monocytes in healthy volunteers given the bacille Calmette–Guérin (BCG) vaccine showed an increased and sustained production of pro-inflammatory cytokines and activation markers in response to various bacterial and fungal pathogens [47]. The polysaccharide β-glucan has also been shown to induce trained immunity in a process that may be dependent on the accumulation of mevalonate [48,49].

In summary, we demonstrated that food-derived polysaccharides such as CPP can enhance the antigen presentation capacity of innate immune cells. Given that influenza vaccine efficacy is dependent on the host’s ability to rapidly recruit competent immune cells and generate long-lasting immunological memory, food-based polysaccharides may be a safe but effective means of boosting vaccine-mediated protection in immunocompromised individuals.

## Figures and Tables

**Figure 1 nutrients-12-02740-f001:**
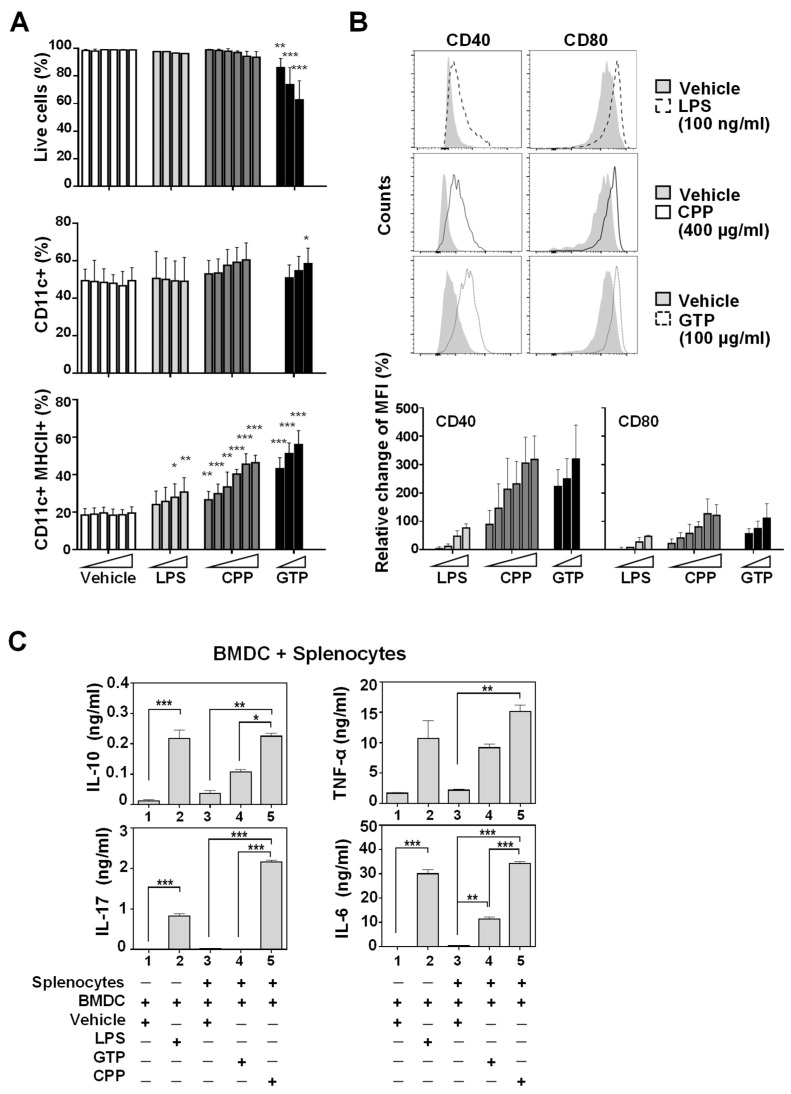
Food-derived polysaccharides carrot pomace polysaccharide (CPP) and green tangerine polysaccharide (GTP) promoted the maturation and activation of bone marrow-derived dendritic cells (BMDCs). (**A**,**B**) GM-CSF-derived BMDC cells were treated with carrot pomace polysaccharides (CPP) or green tangerine polysaccharides (GTP), lipopolysaccharide (LPS), or vehicle at the following concentrations for 24 h and analyzed by flow cytometry (*n*= 3–5); LPS: 0.1, 1, 10, 100 ng/mL; CPP: 100, 200, 400, 600, 800, 1000 µg/mL; GTP: 50, 100, 200 µg/mL. Relative mean fluorescence intensity (MFI) (%) = (polysaccharides-treated sample MFI–vehicle MFI)*/*vehicle MFI * 100. (**c**) BMDCs from C57BL/6 mice were treated with vehicle, CPP (1000 µg/mL), or GTP (100 µg/mL) for 24 h, and co-cultured with BALB/c splenocytes for 3 days. lane1: BMDC + vehicle, lane2: BMDC + LPS, lane3: BMDC + splenocytes + vehicle, lane 4: BMDC + splenocytes + GTP, lane 5: BMDC + splenocytes + CPP. Cytokine levels in mixed lymphocytes’ culture supernatants were quantified with the BD CBA mouse cytokine kit (*n* = 2). Statistical difference by unpaired t-test relative to vehicle (**A**,**B**) or one-way ANOVA with Tukey’s post hoc test (**C**). * *p* < 0.05, ** *p* < 0.01, *** *p* < 0.001.

**Figure 2 nutrients-12-02740-f002:**
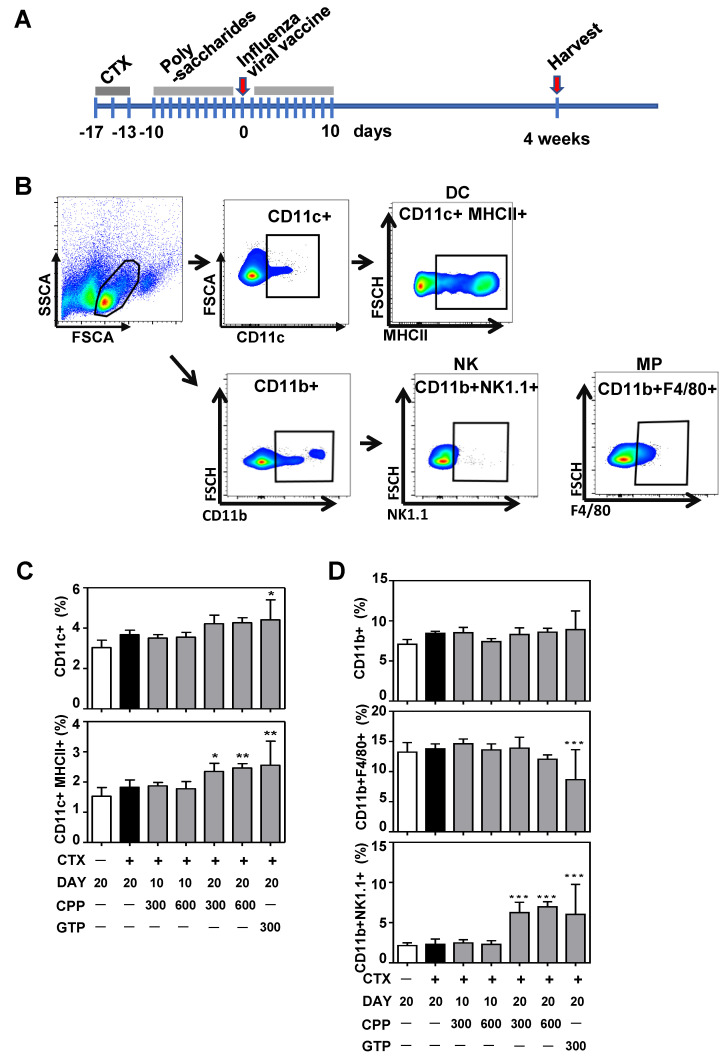
CPP increased the frequency of DCs and NK cells in the splenocytes of vaccinated mice. (**A**) Schematic diagram of the animal experiments. Mice were injected intraperitoneally (IP) with cyclophosphoamide (CTX) and administered an inactivated influenza vaccine. Polysaccharide solutions or vehicle were orally administered either 10 d before vaccination or 10 d both before and after vaccination (*n* = 8). (**B**) Flow cytometry scheme for quantifying dendritic cells (DCs), natural killer cells (NK), and macrophages (MP) in splenocytes isolated from the mice. Frequency of (**C**) DC-enriched (CD11c+ and CD11c+MHCII+) and (**D**) CD11b+ or NK cell-enriched (CD11b+NK1.1+) populations isolated from mice treated at the indicated dose and duration. *n* = 8. Normal distributions of the data were confirmed using GraphPad Prism. Statistical difference to the CTX only control (black bar) by one-way ANOVA with Tukey’s post hoc test. * *p* < 0.05, ** *p* < 0.01, *** *p* < 0.001.

**Figure 3 nutrients-12-02740-f003:**
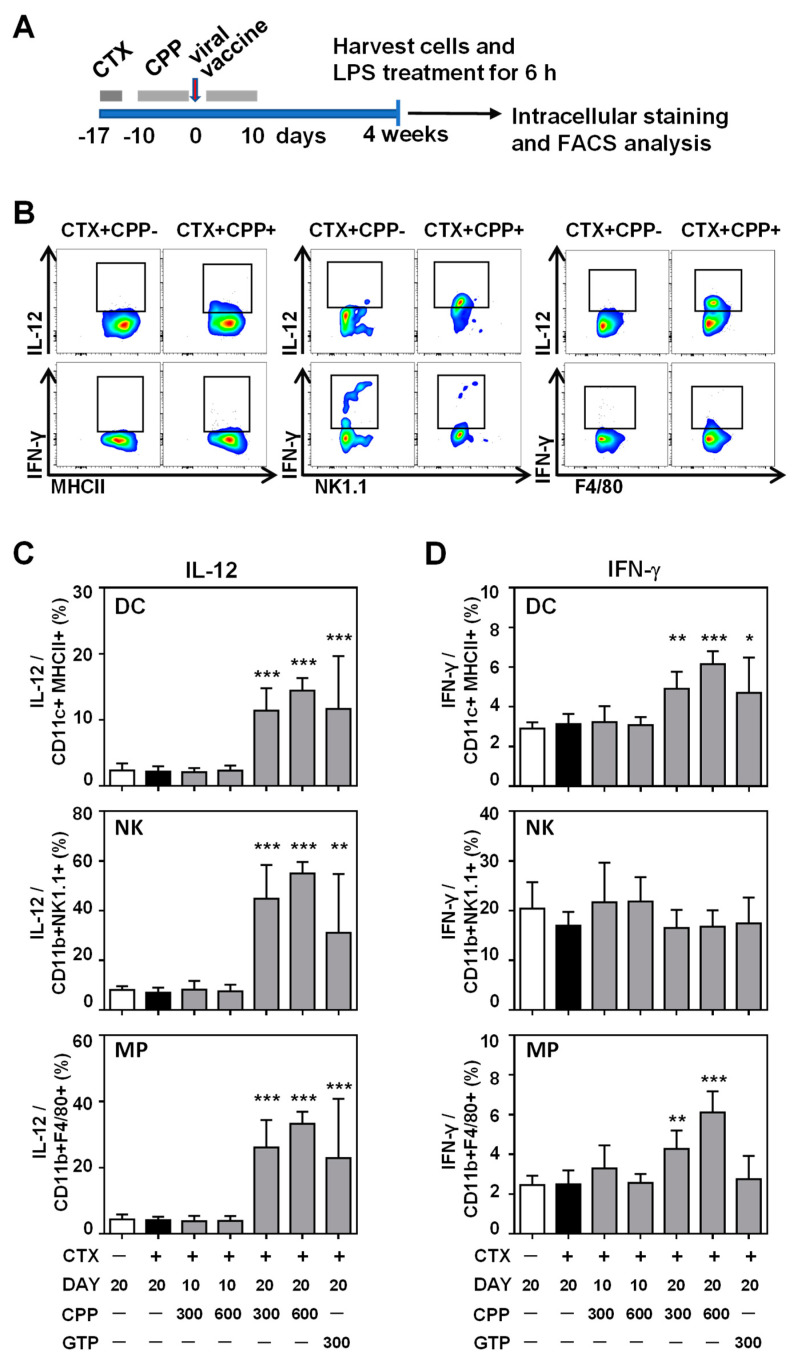
CPP treatment increased the frequency of IL-12+ innate immune cells. (**A**) Schematic diagram of ex vivo experiment. Mice were treated as describe in Figure 2, and the harvested splenocytes were isolated, stimulated with LPS for 6 h, and subjected to cell staining and flow cytometry (*n* = 8). (**B**) Representative flow chart of the CTX-treated samples with or without CPP. Flow cytometry of (**C**) IL-12+ or (**D**) IFN*γ*+ cells in the following cell populations: dendritic cell-enriched CD11c+MHCII+ (DC), NK cell-enriched (NK), and macrophage-enriched CD11b+F4/80+ (MP). Mouse splenocytes were analyzed after stimulation with LPS for 6 h. *n* = 8. Statistical difference by one-way ANOVA with Tukey’s post hoc test. **p* < 0.05, ***p* < 0.01, ****p* < 0.001.

**Figure 4 nutrients-12-02740-f004:**
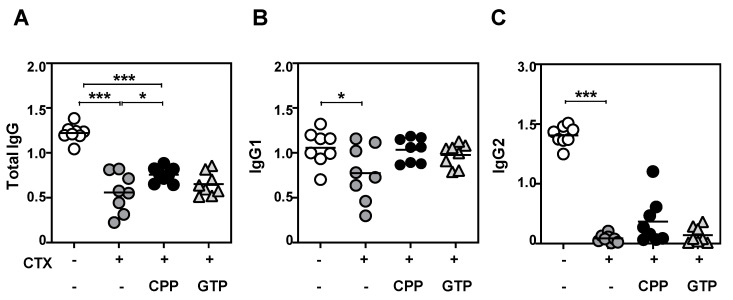
CPP treatment enhanced the antibody response to an influenza virus challenge. Influenza serum antibody titers were measured by total IgG (**A**), IgG1 (**B**), or IgG2 (**C**) ELISA. Mice were administered cyclophosphamide (CTX) and CPP (300 mg/kg), green tangerine polysaccharide (GTP, 300 mg/kg) as a positive control, or vehicle as a negative control. *n* = 8. Statistical difference by one-way ANOVA with Tukey’s post hoc test. * *p* < 0.05, *** *p* < 0.001.

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
