# Peer review of "Carrot Pomace Polysaccharide (CPP) Improves Influenza Vaccine Efficacy in Immunosuppressed Mice via Dendritic Cell Activation"

_nutrients, 2020, doi:10.3390/nu12092740_

Round 1
Reviewer 1 Report
In this article, the authors aimed to find out the efficacy of carrot pomace polysaccharide (CPP) as a vaccine adjuvant. This manuscript provides an essential suggestion, but the experimental concept was not well designed.
Major comments:
1.
The authors showed a significant increase of IL-10, IL-17, and IL-6 in BMDC and splenocytes in Figure 1. Why the authors mixed the strain of the mouse? I think MHC haplotype is different between them. Is that only produced by BMDCs? Is that interact with splenocytes via MHC or cytokines? I encourage the authors to show the cytokine production only from BMDCs or splenocytes. Which cell is the source of these cytokines? IL-17 was produced from Th17 in the spleen? IL-6 was produced from BMDCs? Why did the authors not show the IL-12, which will be valuable in the later, in this experiment?
2.
In Figure 2, the authors tried to show the effect of CPP. Although it was not mentioned, I think the white bar was placed as a negative control. However, the author did not prove the non-treated (CPP negative, GTP negative) of day ten and day 20. I think authors should compare the difference between the CPP positive vs. CPP negative on day 20 with both CTX added. For example, treating only with CTX for 20 days may increase some markers.
3.
In Figure 3, the same comparison was conducted as same as Figure 2. The authors should show whether the CTX concentration in the medium was kept until day 20 or not.
4.
The CPP does not affect ten days incubation, whereas that has a high impact in 20 days incubation in Figure 3. Why is this happen?
5.
The authors showed the oral administration of CPP could alter the vaccine efficacy. The authors need to show whether the CPP administrated via orally is not digested in the peptic juice.
Minor comments:
1.
In Figure 2C, the authors tried to show a significant increase in a group (CTX+, DAY 20, CPP 300, CTP -). The authors should indicate which group was compared as a control in the figure legend.
2.
How much amount of CPP was contained in a carrot?
Author Response
Hello. Please see the attachment.

Reviewer 2 Report
- In Fig 1A, the CPP or GTP was only added for 24h after GM-CSF derived differentiation, not during GM-CSF drived differentiation, therefore it can not be concluded that CPP had an effect on differentiation like what the figure/section title or the main text suggests. Also, it should be open circle for the bottom figure of Fig 1A.
- For Fig 1C, is there an additive effect if adding LPS and CPP together?
- In Fig 2B, flow chart, it should be MHC not MNC
- Have the authors tested antibodies in BALF?
- For Fig3A-B, please add flow chart examples.
Author Response
Hello. Please see the attachment.

Reviewer 3 Report
Summary: In the manuscript entitled, "Carrot pomace polysaccharide (CPP) improves influenza vaccine efficacy in immunosuppressed mice via dendritic cell activation", the authors address the outstanding question of how a polysaccharide derived from food (i.e., carrots) impacts innate immune responses to influenza vaccination. The authors use cellular immunology in vitro assays with mouse cells and in vivo vaccine trials in mice with chemically-induced immune suppression to address this question. While eth data were interesting, enthusiasm was reduced by inconsistencies in reagent concentrations between experiments and eth lack of controls to assess the effects of CTX alone. My detailed comments are provided below.
Figure 1. The data with GTP are limited and not informative for comparisons with CPP, because different concentrations are used and data are missing in dose escalation experiments. Thus, the GTP data is not helpful for interpreting the CPP data. Additional data points for GTP should be provided (gating on live cells) in Figure 1 A & B to provide an adequate positive control or another positive control, such as LPS, should be included to better assess the extent of dendritic cell differentiation/ maturation. This control has been included in Figure 1 C, but it is unclear why it was not included in Figure 1 A & B. In Figure 1 C, the differences in cytokine responses could be explained by differences in concentration between polysaccharides. The concentration of 1000 ug/ml is not physiological, thus the response may be an artifact of the high concentration. To make comparisons with GTP, lower and equivalent concentrations (e.g. 300 ug/ml) would provide a more valid comparison, especially if GTP is the positive control.
Figure 2. It is has been shown previously that CTX treatment alone can enhance immune responses by depleting regulatory T cells (CD4+CD25+). Thus, data from CTX-treated mice without CPP or GTP at day 10 and day 20 should be included. Also evidence should be provided to validate that the CTX treatment was effective in suppressing immunity in this experiment. For this reason the inclusion of data from vaccinated mice without CTX treatment are important for comparison. It is also unclear why different concentrations of CPP were used (300ug/ml and 600 ug/ml). Justification was not provided. Evidence that these reagents are free of LPS should be provided since the CPP and GTP preparations were made using bacteria-derived enzymes. The low frequencies ( low %) of events warrant non-parametric statistical analyses since the data likely do not have normal distribution. Also analyses of geometric means of MFI of the populations of interest should be used for more accurate comparisons between groups.
Figure 3. Again like Figure 2, it is has been shown previously that CTX treatment alone can enhance immune responses by depleting regulatory T cells (CD4+CD25+). Thus, data from CTX-treated mice without CPP or GTP at day 10 and day 20 should be included.
Figure 4. While these data are convincing and highlight the importance of this work, the authors should indicate whether or not there is a statistically significant difference between the Total IgG levels between mice without CTX treatment and CPP/CTX treated mice. This outcome should be included in the discussion.
Author Response
Hello. Please see the attachment.

Reviewer 4 Report
The manuscript entitle” Carrot pomace polysaccharide (CPP) improves influenza vaccine efficacy in immunosuppressed mice via dendritic cell activation” described the efficacy of oral and i.p. administration of CPP on the dendritic cells in a mouse model. There are several issues that need to straighten out.
- Is the endotoxin test data of the CPP available? The authors may need to include this data in the supplemental material to certify the activation of DCs is not from something else.
- Why do the authors use the bone-marrow-derived DCs instead of the cDCs in the spleen or LNs of the mice in Figure 1? The weight of the spleen post-vaccination could also be provided in the figure or in the supplemental data.
- In Figure 2., the frequency of DCs, NK cells, and macrophages in the CD45+immunes cells should be given. The staining panel may need to include the anti-CD45 antibody.
- The method of the intracellular staining should be given, i.e. the IL-12+ cell staining, etc. The supplemental data can not be found from the website.
Author Response
Hello. Please see the attachment.

Round 2
Reviewer 1 Report
The authors addressed most of the points I raised before. However, as I mentioned in comments 1, the authors did not respond the why the authors mixed the strain. My point was that why the MHC mismatch is preferable in this situation? I still think that the authors should show the data with MHC matched data.
Author Response
Hello. Please see the attachment, Thank you

Reviewer 3 Report
The concerns I raised with Figure 1 controls were not addressed adequately. An appropriate positive control to enable a more extensive dose response curve should be included. LPS was used in Figure 1C but not in Figure 1A or !B. Since the positive control that was selected is cytotoxic, a more appropriate control should be used for this experiment.
Author Response
Hello. Please see the attachment. Thank you.

Reviewer 4 Report
The review is still not sure about how much connection between the ex vivo bone marrow derived DC data to and the in vivo vaccine data. The cDCs could be different with the BMDCs in many aspects. If using the CD11c column and magnetic beads, each mouse spleen could give 40-50 k cDCs. The author may consider to pool several mice cells together to do the ex vivo test. Sorting is a even better method to get cDCs.
Author Response

(The authors gave the same response as above.)
